# Development of an assay of plasma neurofilament light chain utilizing immunomagnetic reduction technology

Huei-Chun Liu[1], Wei-Che Lin[2], Ming-Jang Chiu[3], Cheng-Hsien Lu[4], Chin-Yi Lin[1], Shieh-Yueh Yang[1,5]*

1 MagQu Co., Ltd., New Taipei City, Taiwan, 2 Department of Diagnostic Radiology, Kaohsiung Chang Gung Memorial Hospital and Chang Gung University College of Medicine, Kaohsiung, Taiwan, 3 Department of Neurology, National Taiwan University Hospital, Taipei, Taiwan, 4 Department of Diagnostic Neurology, Kaohsiung Chang Gung Memorial Hospital and Chang Gung University College of Medicine, Kaohsiung, Taiwan, 5 MagQu LLC, Surprise, Arizona, United States of America

* syyang@magqu.com

**Data Availability Statement:** All relevant data are within the manuscript and its Supporting Information files.

## Abstract

Axonal damage leads to the release of neurofilament light chain (NFL), which enters the CSF or blood. In this work, an assay kit for plasma NFL utilizing immunomagnetic reduction (IMR) was developed. Antibodies against NFL were immobilized on magnetic nanoparticles to develop an IMR NFL kit. The preclinical properties, such as the standard curve, limit of detection (LoD), and dynamic range, were characterized. Thirty-one normal controls (NC), fifty-two patients with Parkinson's disease (PD) or PD dementia (PDD) and thirty-one patients with Alzheimer's disease (AD) were enrolled in the study evaluating the plasma NFL assay using an IMR kit. T-tests and receiver operating characteristic (ROC) curve analysis were performed to investigate the capability for discrimination among the clinical groups according to plasma NFL levels. The LoD of the NFL assay using the IMR kit was found to be 0.18 fg/ml. The dynamic range of the NFL assay reached 1000 pg/ml. The NC group showed a plasma NFL level of 7.70 ± 4.00 pg/ml, which is significantly lower than that of the PD/PDD (15.85 ± 7.82 pg/ml, $p < 0.001$) and AD (19.24 ± 8.99 pg/ml, $p < 0.001$) groups. A significant difference in plasma NFL levels was determined between the PD and AD groups ($p < 0.01$). Through ROC curve analysis, the cut-off value of the plasma NFL concentration for differentiating NCs from dementia patients (AD and PD/PDD) was found to be 12.71 pg/ml, with a clinical sensitivity and specificity of 73.5% and 90.3%, respectively. The AUC was 0.868. Furthermore, the cut-off value of the plasma NFL concentration for discriminating AD from PD/PDD was found to be 18.02 pg/ml, with a clinical sensitivity and specificity of 61.3% and 65.4%, respectively. The AUC was 0.630. An ultrasensitive assay for measuring plasma NFL utilizing IMR technology was developed. Clear differences in plasma NFL concentrations were observed among NCs and PD and AD patients. These results imply that the determination of plasma NFL is promising not only for screening dementia but also for differential diagnosis.

**Funding:** MagQu Co., Ltd provided funding for this study in the form of salaries for Huei-Chun Liu, Chin-Yi Lin and Shieh-Yueh Yang. MagQu Co., Ltd did not have any additional role in the study design, data collection and analysis, decision to publish, or preparation of the manuscript. The specific roles of these authors are articulated in the 'Author contributions' section.

**Competing interests:** Huei-Chun Liu, Chin-Yi Lin and Shieh-Yueh Yang are employees at MagQu Co., Ltd. Shieh-Yueh Yang is also an employee of MagQu LLC. Shieh-Yueh Yang owns stock in MagQu Co., Ltd. This does not alter our adherence to PLOS ONE policies on sharing data and materials. There are no patents, products in development or marketed products associated with this research to declare.

## Introduction

Neurofilament proteins are major constituents of the neuronal cytoskeleton. There are three subunits of neurofilament proteins: the light (NFL), the medium (NFM) and the heavy (NFH) chains. NFL is a putative biomarker of subcortical large-caliber axonal damage. Elevation of NFL levels in body fluid is relevant to not only brain atrophy but also brain diseases [1]. For example, for subjects carrying the MAPT, GRN, or C90rf72 genotypes, the concentration of NFL in cerebrospinal fluid (CSF) was increased after the onset of frontotemporal dementia (FTD) [2]. Thus, the level of NFL in CSF is a promising biomarker for diagnosing semantic dementia [3]. In addition to FTD, CSF NFL was validated as a screening biomarker for inflammatory disease [4], Creutzfeldt-Jakob disease (CJD) [5], acute neuronal ischemia [6], Alzheimer's disease (AD) [1,7], Parkinson's disease (PD) [8], multiple sclerosis (MS) [9] and traumatic brain injury [10,11].

CSF NFL has been validated to be useful for differentiating various types of neurodegenerative diseases. For example, CSF NFL levels were higher in patients with FTD than in early-onset Alzheimer's disease patients [12]. CSF NFL was proposed as a biomarker to discriminate CJD from AD, FTD or DLB and to differentiate atypical cases from typical patients for CJD and FTD [13]. Therefore, CSF NFL may have an impact on clinical applications for improving diagnostic accuracy for dementia.

Lumbar puncture is necessary for sampling CSF. There are many uncomfortable side effects of lumbar puncture, so it is not easy to perform the CSF NFL assay in clinical practice. Instead of CSF, blood may serve as a promising alternative. However, the concentrations of NFL in blood are extremely low and are hardly detectable using traditional assays. With the development of ultrasensitive assay technologies [14,15], it has become possible to precisely assay NFL in human plasma. Previously reported results show a significant correlation between plasma and CSF NFL concentrations ($\rho > 0.5$, $p < 0.001$) [16–18]. This demonstrates the possibility of evaluating dementia by measuring plasma NFL rather than CSF NFL.

Several independent studies have revealed that the levels of plasma NFL are significantly elevated in patients with mild cognitive impairment (MCI) or AD compared to those in normal controls (NC) [18–20]. The AUC (area under receiver operating characteristic curve) value that discriminates MCI/AD from NC according to plasma NFL is higher than 0.8. Increased NFL levels in plasma were associated with reduced Mini-Mental State Examination (MMSE) ($r < -0.3$, $p < 0.001$) [18–20], hippocampal volume, and thickness in cortex regions in AD [18]. Higher levels of plasma NFL at baseline predicted accelerated declines in these measurements [17,18]. However, plasma NFL levels are not able to predict conversion to AD in MCI [18,19].

A recently published paper shows the feasibility of differentiating PD from NC according to plasma NFL (AUC > 0.7) [21]. PD patients at advanced Hoehn-Yohr stages had increased levels of plasma NFL ($p < 0.001$). Detailed analysis shows that the plasma NFL concentration is modestly correlated with the UPDRS part III ($r = 0.42$, $p < 0.001$). A longitudinal study revealed that plasma NFL was able to predict declines in movement and cognition in PD ($p < 0.05$) [21]. Furthermore, the level of plasma NFL could discriminate atypical Parkinsonism syndrome from typical PD (AUC > 0.8) [16,21].

These published results demonstrate the possible clinical impacts of the diagnosis of AD or PD using plasma NFL. However, to be approved as a regular medical device for clinical use, the preclinical performance of reagents used for assaying NFL must be characterized according to guidelines issued by Clinical and Laboratory Standards Institute (CLSI). Such characterizations have been truthfully reported for the NFL reagents used in published papers. In this work, the preclinical performance of an NFL reagent used with immunomagnetic reduction is

explored. All measurements were conducted according to CLSI guidelines EP5- A3, EP7-A2, EP17-A2, and C28-A2.

## Materials and methods

### Synthesis of NFL reagent

In the IMR assay, the reagent consisted of antibody-functionalized $Fe_3O_4$ magnetic nanoparticles dispersed in phosphate-buffered saline (PBS) (1x). To achieve the specific association between magnetic nanoparticles and NFL, antibodies (sc20011, Santa Cruz) against NFL were covalently bound to dextran, which acted as an interface between the antibody and the $Fe_3O_4$ cores of the nanoparticles. A detailed description of the binding of the antibody to dextran is given in [22]. The mean diameter of the magnetic nanoparticles was 53 nm, as measured by laser dynamic scattering. The concentration of the NFL reagent was 10 mg Fe/ml. The reagent was stored at 4 ˚C.

### IMR measurement

A mixture of 60 μl of NFL reagent and 60 μl of sample was used for the IMR measurement. A superconducting quantum interference device-based alternative-current magnetic susceptometer (XacPro-S, MagQu) was used to detect the IMR signal of a sample [23–26]. During detection of the IMR signal of a sample, two control solutions were used. One was blank, i.e., PBS solution, and the other contained 50 pg/ml NFL (Ab224840, Abcam) in PBS solution.

### Exploration of preclinical performance via assay of NFL

All experiments were designed and conducted according to the global standards EP5- A3, EP7-A2, EP17-A2, and C28-A2 described by the Clinical & Laboratory Standards Institute (CLSI). Thus, the standard curve, detection limit, assay linearity, dilution recovery range, assay reproducibility, spike recovery, reagent stability, and interference tests were investigated.

### Enrollment of subjects

Subjects were enrolled at National Taiwan University Hospital and Kaohsiung Chang Gung Memorial Hospital with the approval of the ethics committees of both hospitals. Every participant provided written informed consent. Enrolled subjects were categorized into groups of normal controls (NC) and patients with Parkinson's disease (PD), or PD dementia (PDD), and Alzheimer's disease (AD) according to the inclusion and exclusion criteria listed in Table 1. PD, PDD and AD were combined as patient group. All experimental protocols with human samples were approved by the ethics committees at both hospitals.

### Preparation of plasma samples

The blood draw was performed using a 9 ml or 6 ml K3 EDTA lavender-top tube, followed by centrifugation at 1500–2500 g at room temperature for 15 minutes. Five hundred microliters of plasma was aliquoted into each 0.5-ml cryotube and stored at -20 ˚C or -80 ˚C. The plasma was required to be frozen no later than 3 hours after the blood draw. Each frozen plasma aliquot was placed in wet ice and then brought to room temperature prior to the IMR measurement.

**Table 1. Exclusion and inclusion criteria used for recruiting normal controls (NC) and subjects with Alzheimer's disease (AD) and Parkinson's disease in this study.**

| Group | Inclusion criteria | Exclusion criteria |
|---|---|---|
| NC | 1. Education: at least primary school<br>2. Age > 50 years<br>3. Body weight ≥ 40 kg<br>4. CDR* = 0<br>5. MMSE++ ≥ 26 | 1. Subjects with cranial metallic implants, cardiac pacemakers or claustrophobia.<br>2. Previous diagnosis of MCI or dementia<br>3. Significant history of depression<br>4. Geriatric Depression Scale > 8 |
| PD/PDD | 1. Subjects must have symptoms of bradykinesia and at least one of the following: muscular rigidity, resting tremor (4–6 Hz), or postural instability unrelated to primary visual, cerebellar, vestibular or proprioceptive dysfunction [27].<br>2. Three or more of the following symptoms: unilateral onset, resting tremor, progressive disorder, persistent asymmetry most affecting the side of onset, excellent response to levodopa, severe levodopa-induced chorea, levodopa response for over 5 years, and clinical course of over 10 years.<br>3. MOCA# score greater than 26 for PD with normal cognition<br>4. MOCA score less than 21 for PD with dementia | 1. Significant history of depression<br>2. History of repeated strokes with stepwise progression, repeated head injury, antipsychotic or dopamine-depleting drugs, definite encephalitis and/or oculogyric crises on no drug treatment, negative response to large doses of levodopa (if malabsorption was excluded), strictly unilateral features after 3 years, other neurological features (supranuclear gaze palsy, cerebellar signs, early severe autonomic involvement, Babinski sign, early severe dementia with disturbances of language, memory or praxis), exposure to a known neurotoxin, or presence of cerebral tumor or communicating hydrocephalus according to neuroimaging. |
| AD | 1. Subjects must meet the 2011 NIA-AA diagnostic guidelines for probable AD dementia [28].<br>2. Subjects must have MMSE scores between 10 and 22 and CDR = 0.5 or 1. | 1. Subjects with cranial metallic implants, cardiac pacemakers or claustrophobia.<br>2. Significant history of depression<br>3. Geriatric Depression Scale > 8 |

*CDR: Clinical Dementia Ranking

++MMSE: Mini-Mental State Examination

#MOCA: Montreal Cognitive Assessment

## Statistical methods

The continuous variables for each measurement are presented as the means ± standard deviations. The data of NFL concentrations were checked for normality using the Kolmogorov-Smirnov test and the result showed suggested a violation in the normality assumption ($p = 0.033$; data not shown). Therefore, the comparison of NFL concentrations between groups (i.e., normal control vs. disease groups; PD/PDD vs. AD groups) was made using the Mann-Whitney U-test. To clarify the discrimination between two of PD/PDD, AD, patient and NC, receiver operating characteristic (ROC) curve analysis was conducted. The optimal cutoff of NFL concentrations was determined by the Youden index. The confidence interval of area under the curve, sensitivity and specificity was calculated using the DeLong's nonparametric method. At last, the relationship between spiked NFL and measure NFL (both were log10-transformed) was assessed using the Pearson's correlation. A two-sided P value less than 0.5 was considered statistically significant. Analyses were done using GraphPad Prism 5 and Med-Calc version 13.

## Results

### Hook effect on the assay

The NFL concentration-dependent IMR signal was investigated. Phosphate-buffered saline (PBS) solutions spiked with NFL (Ab224840, Abcam) at various concentrations were used for the IMR measurements. The concentrations of the spiked NFL samples, denoted as $\phi_{NFL}$,

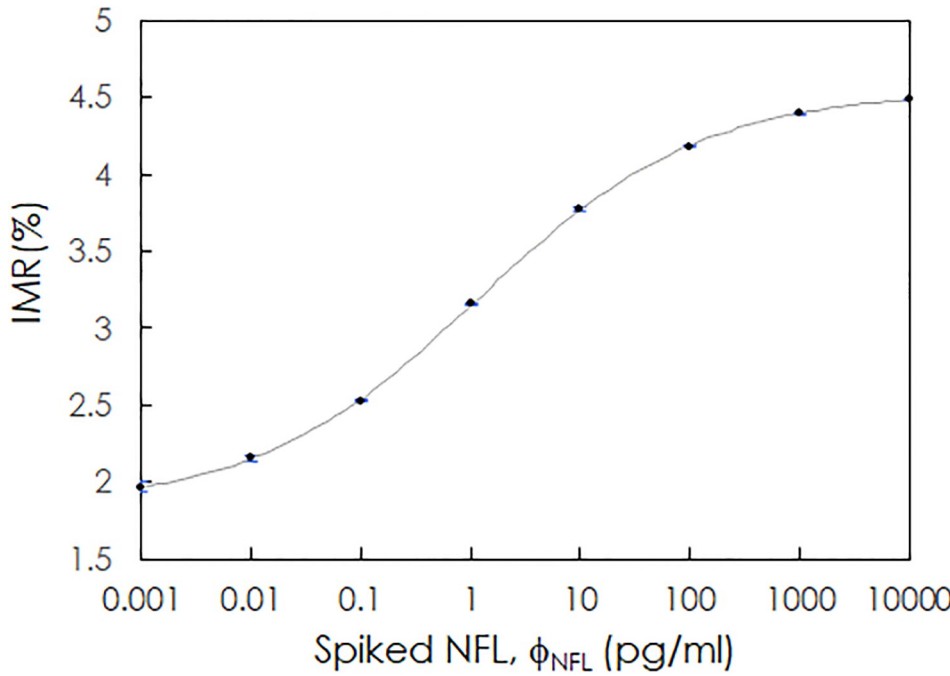

**Fig 1. Relationship between the IMR signal, IMR(%), and the spiked NFL concentrations.** The gray solid line is the fitted curve determined with Eq (1).

ranged from 0.001 to 10000 pg/ml. The measured IMR signal, denoted as IMR(%), was 1.967 for 0.001 pg/ml NFL and increased to 4.487 for 10000 pg/ml NFL. The relationship between IMR(%) and $\phi_{NFL}$ is plotted in Fig 1. The data points were well fitted to the logistic function

$$\text{IMR}(\%) = \frac{A - B}{1 + (\frac{\phi_{NFL}}{\phi_o})^{\gamma}} + B, \tag{1}$$

where A, B, $\phi_o$ and $\gamma$ were fitting parameters and were found to be 1.840, 4.538, 1.166 and 0.428, respectively. $\Phi_{NFL}$ is the spiked NFL concentration in PBS. The fitted logistic curve is plotted as a gray line in Fig 1. The data point at 10000 pg/ml NFL still lies on the fitted line. This implies that the Hook effect does not occur at 10000 pg/ml NFL.

## Assay detection limit

The limit of blank (LoB) and the limit of detection (LoD) were investigated according to the guidelines for evaluating the detection capacity of clinical laboratory measurement procedures described in CLSI EP17-A2. The LoB was obtained by determining the appropriate percentile (p) value of the ranked measured concentrations for the blank samples, which was p = 0.95 in this case:

$$\text{LoB} = \text{Results at position } [0.95 \text{ x } N_B + 0.5], \tag{2}$$

where $N_B$ = 60 ($N_B$ is the number of trials) in the current case. Eq (2) becomes

$$\text{LoB} = \text{Results at position } 57.5 \tag{3}$$

**Table 2. Ranked list of the 60 measured NFL concentrations for PBS samples not spiked with NFL using the IMR NFL reagent.**

| Rank | Measured concentration (fg/ml) | Rank | Measured concentration (fg/ml) |
|------|-------------------------------|------|-------------------------------|
| 1 | -0.03 | 31 | 0.00 |
| 2 | -0.03 | 32 | 0.00 |
| 3 | -0.02 | 33 | 0.00 |
| 4 | -0.02 | 34 | 0.00 |
| 5 | -0.02 | 35 | 0.00 |
| 6 | -0.02 | 36 | 0.00 |
| 7 | -0.02 | 37 | 0.00 |
| 8 | -0.02 | 38 | 0.00 |
| 9 | -0.02 | 39 | 0.00 |
| 10 | -0.02 | 40 | 0.00 |
| 11 | -0.02 | 41 | 0.00 |
| 12 | -0.01 | 42 | 0.00 |
| 13 | -0.01 | 43 | 0.00 |
| 14 | -0.01 | 44 | 0.00 |
| 15 | -0.01 | 45 | 0.00 |
| 16 | -0.01 | 46 | 0.00 |
| 17 | -0.01 | 47 | 0.00 |
| 18 | -0.01 | 48 | 0.00 |
| 19 | -0.01 | 49 | 0.00 |
| 20 | -0.01 | 50 | 0.00 |
| 21 | -0.01 | 51 | 0.00 |
| 22 | -0.01 | 52 | 0.00 |
| 23 | -0.01 | 53 | 0.00 |
| 24 | -0.01 | 54 | 0.00 |
| 25 | -0.01 | 55 | 0.00 |
| 26 | -0.01 | 56 | 0.00 |
| 27 | 0.00 | 57 | 0.00 |
| 28 | 0.00 | 58 | 0.00 |
| 29 | 0.00 | 59 | 0.00 |
| 30 | 0.00 | 60 | 0.00 |

The LoB was calculated by performing linear interpolation using the 57th- and 58th-ranked measured concentrations. Table 2 lists the 60 ranked measured concentrations for the PBS samples (blank samples) without NFL. The 57.5th-ranked measured concentration was 0.000 fg/ml. Therefore, the LoB used for assaying NFL with the IMR NFL reagent was 0.000 fg/ml.

The limit of detection (LoD) was calculated with the equation

$$\mathrm{LoD} = \mathrm{LoB} + 1.65\sigma_\mathrm{S}, \tag{4}$$

where $\sigma_\mathrm{S}$ denotes the standard deviation of the measured NFL concentrations of the NFL solutions spiked with a fixed NFL concentration (e.g., 1.0 fg/ml in the current case) in PBS. Table 3 lists the measured NFL concentrations of the 60 NFL solutions. The mean value of the 60 measured concentrations was 1.05 fg/ml. The $\sigma_\mathrm{S}$ of the 60 measured concentrations was 0.11 fg/ml. The LoD for assaying NFL using IMR was 0.18 fg/ml according to Eq (4).

**Table 3. Ranked list of the 60 measured NFL concentrations for PBS samples spiked with 1.0 fg/ml NFL using the IMR NFL reagent.**

| Rank | Measured concentration (fg/ml) | Rank | Measured concentration (fg/ml) |
|---|---|---|---|
| 1 | 0.86 | 31 | 1.06 |
| 2 | 0.90 | 32 | 1.06 |
| 3 | 0.91 | 33 | 1.06 |
| 4 | 0.92 | 34 | 1.06 |
| 5 | 0.92 | 35 | 1.07 |
| 6 | 0.93 | 36 | 1.08 |
| 7 | 0.93 | 37 | 1.08 |
| 8 | 0.93 | 38 | 1.08 |
| 9 | 0.93 | 39 | 1.09 |
| 10 | 0.94 | 40 | 1.09 |
| 11 | 0.94 | 41 | 1.09 |
| 12 | 0.94 | 42 | 1.09 |
| 13 | 0.95 | 43 | 1.09 |
| 14 | 0.95 | 44 | 1.09 |
| 15 | 0.96 | 45 | 1.10 |
| 16 | 0.96 | 46 | 1.10 |
| 17 | 0.96 | 47 | 1.10 |
| 18 | 0.97 | 48 | 1.11 |
| 19 | 0.99 | 49 | 1.11 |
| 20 | 0.99 | 50 | 1.12 |
| 21 | 1.00 | 51 | 1.13 |
| 22 | 1.01 | 52 | 1.14 |
| 23 | 1.01 | 53 | 1.14 |
| 24 | 1.03 | 54 | 1.17 |
| 25 | 1.05 | 55 | 1.20 |
| 26 | 1.05 | 56 | 1.20 |
| 27 | 1.05 | 57 | 1.21 |
| 28 | 1.05 | 58 | 1.31 |
| 29 | 1.05 | 59 | 1.38 |
| 30 | 1.05 | 60 | 1.41 |

## Assay linearity

The linear range of the NFL assay is defined according to the NFL concentrations that reflect the proportionality between the measured NFL concentration $\phi_{\text{NFL,m}}$ and the spiked NFL concentration $\phi_{\text{NFL}}$. The proportional coefficient should be between 0.9 and 1.1, and the coefficient of determination $R^2$ for the $\phi_{\text{NFL,m}}$-$\phi_{\text{NFL}}$ curve should be higher than 0.95. To investigate the assay linearity, the detected IMR (%) values in Fig 1 were converted to the measured NFL concentrations $\phi_{\text{NFL,m}}$ via Eq (1). Fig 2 plots the relationship between $\phi_{\text{NFL,m}}$ and $\phi_{\text{NFL}}$. It was found that $\phi_{\text{NFL,m}}$ is proportional to $\phi_{\text{NFL}}$, as expressed by

$$\phi_{\text{NFL,m}} = s\phi_{\text{NFL}} \tag{5}$$

The proportional coefficient s depends on the selected ranges of NFL concentrations. For example, for the selected NFL concentration range from 0.001 to 10000 pg/ml, s and $R^2$ are equal to 1.17 and 0.9998, respectively, according to the plotted gray dotted line in Fig 2. In this

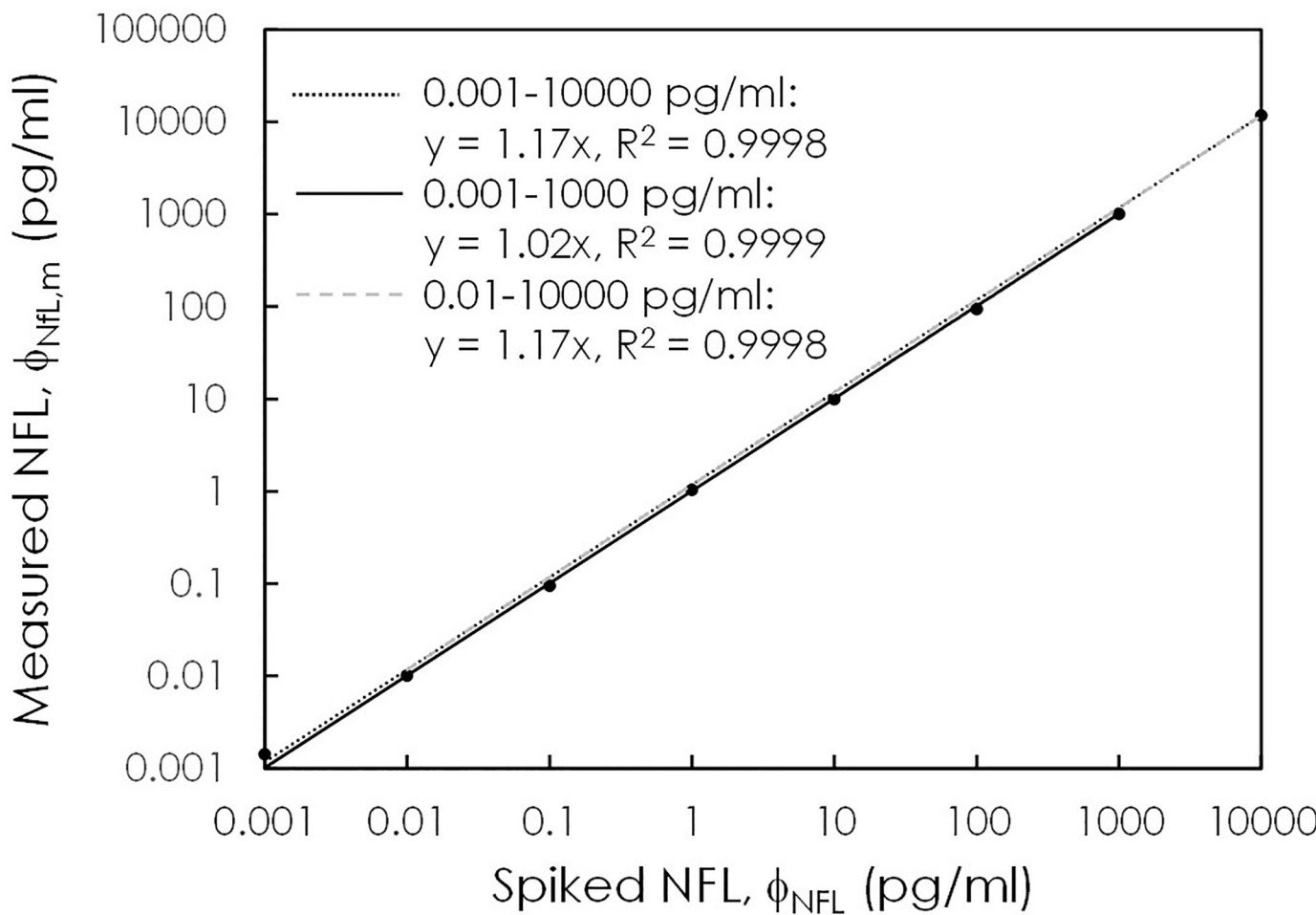

**Fig 2. Relationship between the measured NFL concentrations using IMR and the spiked NFL concentrations.** The solid/dotted/dashed lines show the proportionalities between the measured NFL concentrations and the spiked NFL concentrations according to various ranges of spiked NFL concentrations.

case, the proportional coefficient is higher than 1.1, which does not meet the requirement of the linear range. The data of the highest concentration to the lowest concentration are removed to find s in Eq (5), i.e., 0.001 to 1000 pg/ml or 0.01 to 10000 pg/ml. The results show that s is 1.02 and $R^2$ is 0.9999 once the concentration range is selected to be from 0.001 to 1000 pg/ml, according to the plotted solid line in Fig 2. Therefore, the linear range for the NFL assay using the IMR NFL reagent ranges from 0.001 to 1000 pg/ml.

### Dilution recovery range

The dilution recovery is defined as

$$\text{Dilution recovery} = \frac{\text{measured concentration after dilution}}{\text{expected concentration after dilution}} x 100\% \qquad (6)$$

The acceptable dilution recovery range is from 90% to 110%. In this work, an NFL solution with a measured concentration of 101.87 pg/ml was diluted by a factor of between 2 and 100 with PBS solution. The expected and measured NFL concentrations after dilution are listed in

**Table 4. Dilution factors, expected concentrations, measured concentrations, and dilution recovery of diluted samples used in the tests of the dilution recovery range for assaying NFL using the IMR NFL reagent.**

| Dilution factor | Expected concentration (pg/ml) | Measured concentration (pg/ml) | Dilution recovery |
|---|---|---|---|
| 2X | 50.93 | 52.63 | 103.3% |
| 5X | 20.37 | 20.43 | 100.3% |
| 10X | 10.19 | 10.56 | 103.7% |
| 20X | 5.09 | 5.36 | 105.2% |
| 50X | 2.04 | 2.11 | 103.5% |
| 100X | 1.02 | 1.03 | 101.1% |

Table 4. The dilution recoveries of the 2X- to 100X-diluted samples ranged from 1001.1% to 105.2%, which are within the acceptable dilution recovery range. Hence, the sample used for the IMR NFL assay could be diluted up to 100 times.

## Spiked recovery

A sample with a higher NFL concentration was spiked into another sample with a lower NFL concentration. The NFL concentration of the mixture was assayed with IMR NFL reagent. The spiked recovery was calculated via

$$\text{Spiked recovery} = \frac{\text{measured concentration after mixing}}{\text{expected concentration after mixing}} x 100\% \tag{7}$$

Two plasma samples with original concentrations of 5.39 (plasma sample no. PRA) and 101.09 pg/ml (plasma sample no. PRF) were mixed at various volume ratios, as shown in Table 5. The original concentrations were assayed using IMR NFL reagent. The results in Table 5 reveal that the spiked recovery ranged from 93.4% to 99.1% for the IMR NFL reagent.

## Assay reproducibility

The reproducibility of the NFL assay with IMR was investigated according to CLSI EP5-A3: Approved Guidelines for Evaluation of the Precision Performance of Quantitative Measurement Methods. The NFL solutions were measured precisely in one run using IMR NFL reagent. Two sequential measurements of two duplicate measurements each were regarded as two runs, referred to as Run1 and Run2. Two PBS solutions spiked with different NFL concentrations were used for the tests of reproducibility. The measured NFL concentrations are listed in Table 6 for NFL-PBS sample 1 and in Table 7 for NFL-PBS sample 2. The results show that

**Table 5. Measured NFL concentration using the IMR NFL reagent and the spiked recovery of plasma samples.**

| Plasma sample No. | Volume ratio (PRA: PRF) | Original concentration (pg/ml) | Expected concentration (pg/ml) | Measured concentration (pg/ml) | Spiked recovery |
|---|---|---|---|---|---|
| PRA | - | 5.39 | - | - | - |
| PRB | 80%:20% | - | 24.53 | 22.91 | 93.4% |
| PRC | 60%:40% | - | 43.67 | 41.41 | 94.8% |
| PRD | 40%:60% | - | 62.81 | 62.27 | 99.1% |
| PRE | 20%:80% | - | 81.95 | 80.22 | 97.0% |
| PRF | - | 101.09 | - | - | - |

**Table 6. Measured NFL concentrations (listed in the Result11, Result12, Result21 and Result22 columns) in NFL-PBS sample 1 used for the analysis of the precision and reproducibility of the IMR NFL reagent.**

| | | Run 1 | | | | | | Run 2 | | | | ΔMean | Mean (pg/ml) |
|---|---|---|---|---|---|---|---|---|---|---|---|---|---|
| Day | Date | Result11 (pg/ml) | Result12 (pg/ml) | Mean1 (pg/ml) | ΔResult1 | Day | Date | Result21 (pg/ml) | Result22 (pg/ml) | Mean2 | ΔResult2 | | |
| 1 | 2019/7/24 | 1.04 | 1.00 | 1.02 | 0.002 | 2 | 2019/7/29 | 0.995 | 1.097 | 1.046 | 0.010 | 0.001 | 1.03 |
| 3 | 2019/7/31 | 1.011 | 1.008 | 1.01 | 0.000 | 4 | 2019/8/1 | 1.018 | 1.002 | 1.010 | 0.000 | 0.000 | 1.01 |
| 5 | 2019/8/2 | 0.964 | 1.010 | 0.99 | 0.002 | 6 | 2019/8/6 | 1.000 | 1.020 | 1.010 | 0.000 | 0.001 | 1.00 |
| 7 | 2019/8/7 | 1.018 | 0.992 | 1.01 | 0.001 | 8 | 2019/8/13 | 0.995 | 0.995 | 0.995 | 0.000 | 0.000 | 1.00 |
| 9 | 2019/8/14 | 1.012 | 0.990 | 1.00 | 0.000 | 10 | 2019/8/21 | 0.950 | 0.980 | 0.965 | 0.001 | 0.001 | 0.98 |
| 11 | 2019/8/27 | 1.015 | 1.020 | 1.02 | 0.000 | 12 | 2019/8/30 | 1.010 | 1.010 | 1.010 | 0.000 | 0.000 | 1.01 |
| 13 | 2019/9/3 | 1.010 | 0.972 | 0.99 | 0.001 | 14 | 2019/9/4 | 0.985 | 1.010 | 0.998 | 0.001 | 0.000 | 0.99 |
| 15 | 2019/9/5 | 0.995 | 1.005 | 1.00 | 0.000 | 16 | 2019/9/9 | 0.995 | 1.025 | 1.010 | 0.001 | 0.000 | 1.01 |
| 17 | 2019/9/10 | 1.005 | 1.000 | 1.00 | 0.000 | 18 | 2019/9/12 | 0.970 | 0.975 | 0.973 | 0.000 | 0.001 | 0.99 |
| 19 | 2019/9/17 | 1.005 | 1.010 | 1.01 | 0.000 | 20 | 2019/9/18 | 0.980 | 0.990 | 0.985 | 0.000 | 0.001 | 1.00 |
| Sum | | | | | 0.007 | | | | | | 0.013 | 0.004 | 10.02 |

Mean1 = (Result11+Result12)/2    ΔResult1 = (Result11-Result12)^2
Mean2 = (Result21+Result22)/2    ΔResult2 = (Result21-Result22)^2
ΔMean = (Mean1-Mean2)^2    Mean = (Mean1+Mean2)/2

NFL-PBS sample 1 had a concentration of 1.00 pg/ml, and NFL-PBS sample 2 had a concentration of 100.17 pg/ml. By following the statistical method described in the CLSI EP5-A3 guidelines, the within-lab precision and standard deviations of repeatability for NFL-PBS samples 1 and 2 were calculated and are listed in Table 8. The imprecision (%CV) of the NFL assay using the IMR NFL reagent was less than 10%.

**Table 7. Measured NFL concentrations (listed in the Result11, Result12, Result21 and Result22 columns) in NFL-PBS sample 2 used for the analysis of the precision and reproducibility of the IMR NFL reagent.**

| | | Run1 | | | | | | Run2 | | | | ΔMean | Mean (pg/ml) |
|---|---|---|---|---|---|---|---|---|---|---|---|---|---|
| Day | Date | Result11 (pg/ml) | Result12 (pg/ml) | Mean1 (pg/ml) | ΔResult1 | Day | Date | Result21 (pg/ml) | Result22 (pg/ml) | Mean2 | ΔResult2 | | |
| 1 | 2019/7/24 | 99.51 | 100.31 | 99.91 | 0.640 | 2 | 2019/7/29 | 100.735 | 99.625 | 100.180 | 1.232 | 0.073 | 100.05 |
| 3 | 2019/7/31 | 100.245 | 100.085 | 100.17 | 0.026 | 4 | 2019/8/1 | 99.780 | 99.375 | 99.578 | 0.164 | 0.345 | 99.87 |
| 5 | 2019/8/2 | 99.820 | 99.655 | 99.74 | 0.027 | 6 | 2019/8/6 | 99.865 | 99.710 | 99.788 | 0.024 | 0.002 | 99.76 |
| 7 | 2019/8/7 | 99.975 | 99.975 | 99.98 | 0.000 | 8 | 2019/8/13 | 100.835 | 99.630 | 100.233 | 1.452 | 0.066 | 100.10 |
| 9 | 2019/8/14 | 99.945 | 100.075 | 100.01 | 0.017 | 10 | 2019/8/21 | 100.195 | 99.830 | 100.013 | 0.133 | 0.000 | 100.01 |
| 11 | 2019/8/27 | 100.315 | 100.465 | 100.39 | 0.023 | 12 | 2019/8/30 | 99.615 | 100.390 | 100.003 | 0.601 | 0.150 | 100.20 |
| 13 | 2019/9/3 | 99.915 | 100.035 | 99.98 | 0.014 | 14 | 2019/9/4 | 104.440 | 100.350 | 102.395 | 16.728 | 5.856 | 101.19 |
| 15 | 2019/9/5 | 100.440 | 99.770 | 100.11 | 0.449 | 16 | 2019/9/9 | 100.500 | 100.110 | 100.305 | 0.152 | 0.040 | 100.21 |
| 17 | 2019/9/10 | 99.905 | 99.925 | 99.92 | 0.000 | 18 | 2019/9/12 | 99.570 | 100.485 | 100.028 | 0.837 | 0.013 | 99.97 |
| 19 | 2019/9/17 | 100.250 | 101.895 | 101.07 | 2.706 | 20 | 2019/9/18 | 99.760 | 99.405 | 99.583 | 0.126 | 2.220 | 100.33 |
| Sum | | | | | 3.902 | | | | | | 21.449 | 8.766 | 1001.68 |

Mean1 = (Result11+Result12)/2    ΔResult1 = (Result11-Result12)^2
Mean2 = (Result21+Result22)/2    ΔResult2 = (Result21-Result22)^2
ΔMean = (Mean1-Mean2)^2    Mean = (Mean1+Mean2)/2

**Table 8. Standard deviations of the repeatability and within-lab precision for the assay of NFL concentrations in PBS using the IMR NFL reagent.** The samples used show mean measured NFL concentrations of 1.00 pg/ml and 100.17 pg/ml. The coefficient of variation is the ratio of the standard deviation and the mean of the measured NFL concentrations.

| Material | Mean of measured NFL concentrations | Standard deviation (Coefficient of variation) | |
|---|---|---|---|
| | | Repeatability | Within-lab precision |
| NFL-PBS sample 1 | 1.00 pg/ml | 0.027 pg/ml (2.67%) | 0.019 pg/ml (1.89%) |
| NFL-PBS sample 2 | 100.17 pg/ml | 0.528 pg/ml (0.53%) | 1.313 pg/ml (1.31%) |

## Interference test

Possible false signals during the measurement of NFL concentrations due to molecules referred to as interfering materials in the tested sample were investigated. Table 9 lists all the interfering materials tested in this work. Sample No. 1 was pure PBS solution spiked only with 100 pg/ml NFL without any interfering material. The measured NFL concentration of Sample No. 1 (= 99.35 pg/ml) was used as a reference. The recovery rate of a given sample (Sample No. 2–17) was the ratio of the measured NFL concentration of the given sample to that of the reference sample. The recovery rate ranged from 93.4% to 106.4%, which satisfied the requirement for nonsignificant interference in the NFL assay. The results show that the molecules listed in Table 9, including NF heavy and medium peptides, did not significantly interfere with the NFL assay using the IMR NFL reagent.

## Plasma NFL concentrations in patients with dementia

In this work, thirty-one normal controls (NC), fifty-two patients with Parkinson's disease (PD) or PD dementia (PDD), and thirty-one patients with Alzheimer's disease (AD) were enrolled at National Taiwan University Hospital and Kaohsiung Chang Gung Memorial

**Table 9. Materials and their concentrations used for interference tests for the NFL assay utilizing the NFL reagent with IMR.** The concentration of NFL in each sample was 100 pg/ml. The matrix was PBS solution. The measured NFL concentrations of each sample are listed. Using the NFL concentration of the pure NFL-PBS sample (sample No. 1) as a reference, the recovery rates of the NFL concentrations of the other samples were calculated and are listed in the rightmost column.

| Sample No. | Interfering material | Concentration | Measured NFL concentration (pg/ml) | Recovery rate |
|---|---|---|---|---|
| 1 | None | - | 99.35 | - |
| 2 | Conjugated bilirubin | 600 μg/ml | 96.16 | 96.8% |
| 3 | Hemoglobin | 10000 μg/ml | 105.46 | 106.2% |
| 4 | Intralipid | 30000 μg/ml | 97.03 | 97.7% |
| 5 | Albumin | 60000 μg/ml | 92.77 | 93.4% |
| 6 | Rheumatoid factor | 500 IU/ml | 96.30 | 96.9% |
| 7 | Uric acid | 200 μg/ml | 100.92 | 101.6% |
| 8 | Neurofilament, Heavy peptide | 100 pg/ml | 101.29 | 102.0% |
| 9 | Neurofilament, Medium peptide | 100 pg/ml | 105.69 | 106.4% |
| 10 | Acetylsalicylic acid | 500 μg/ml | 97.06 | 97.7% |
| 11 | Ascorbic acid | 300 μg/ml | 97.35 | 98.0% |
| 12 | Ampicillin sodium | 1000 μg/ml | 101.21 | 101.9% |
| 13 | Quetiapine fumarate | 100 μg/ml | 100.87 | 101.5% |
| 14 | Galantamine hydrobromide | 90 ng/ml | 98.23 | 98.9% |
| 15 | Rivastigmine hydrogen tartrate | 100 ng/ml | 96.36 | 97.0% |
| 16 | Donepezil hydrochloride | 1000 ng/ml | 100.44 | 101.1% |
| 17 | Memantine hydrochloride | 150 ng/ml | 102.75 | 103.4% |

**Table 10. Brief demographic information for the enrolled subjects.**

| Group (n) | NC (31) | Patient (83)* | PD/PDD (52) | AD (31) |
|---|---|---|---|---|
| Female% | 80.0 | 56.5 | 61.7 | 48.0 |
| Age (years) | 60.3 ± 7.2 | 66.8 ± 10.3 | 62.6 ± 9.5 | 74.6 ± 9.6 |
| MMSE | 28.0 ± 1.9 | 23.2 ± 5.3 | 24.8 ± 4.6 | 20.2 ± 5.3 |
| H-Y stage | - | - | 1.96 ± 0.98 | - |
| NFL (pg/ml) | 7.70 ± 4.00 | 17.11 ± 8.39 | 15.85 ± 7.82 | 19.24 ± 8.99 |

*: Patient includes PD/PDD and AD NC: normal control PD: Parkinson's disease

PDD: Parkinson's disease dementia AD: Alzheimer's disease

MMSE: Mini-Mental State Examination

H-Y stage: Hoehn and Yahr stage

Hospital. The demographic information of the subjects is listed in Table 10. NCs had a mean level of 7.70 ± 4.00 pg/ml for plasma NFL. Patient group showed 66.8 ± 10.3 pg/ml for plasma NFL level, which is significantly higher than NCs ($p < 0.001$), as shown in Fig 3(A). A significant difference in the plasma NFL concentrations between AD (19.24 ± 8.99 pg/ml) and PD/PDD (15.85 ± 7.82 pg/ml) ($p < 0.05$) was found.

By using the ultrasensitive and highly specific IMR NFL assay, the clear discrimination of the concentrations of plasma NFL in NCs (7.70 ± 4.00 pg/ml) and PD/PDD patients (15.85 ± 7.82 pg/ml) was achieved ($p < 0.001$). Through the analysis of the receiver operating characteristic (ROC) curve plotted with the black solid line in Fig 3(B), the cut-off value of the plasma NFL concentration used to discriminate PD/PDD patients from NCs was found to be 12.71 pg/ml, which resulted in values of 0.712 and 0.903 for clinical sensitivity and specificity, respectively. The corresponding area under the curve (AUC) was 0.838, as shown in Table 11. These results reveal the feasibility of differentiating PD/PDD patients from normal controls.

In addition to PD/PDD patients, AD patients (19.24 ± 8.99 pg/ml) showed higher levels of plasma NFL than NCs ($p < 0.001$). The ROC curve for differentiating AD patients from NCs using plasma NFL concentrations is plotted with the gray solid line in Fig 3(B). The cut-off value (= 12.04 pg/ml), sensitivity (= 0.838), specificity (= 0.871) and AUC (= 0.919) values are listed in Table 12. Notably, the accuracy of differentiating AD patients from NCs was higher than 85%. By combining PD/PDD and AD into a patient group, discrimination between patient and NCs using plasma NFL concentrations was investigated. The ROC curve is plotted as the black dashed line in Fig 3(B). The cut-off value of the plasma NFL concentration used to discriminate patient from NCs is 12.71 pg/ml, the sensitivity is 0.735, the specificity is 0.903 and the AUC is 0.868. These results demonstrate the feasibility of identifying PD/PDD or AD using plasma NFL concentrations. This was also shown by other clinical studies listed in Table 12 [1,16,20,21,29]. However, differentiation between AD and PD/PDD patients using plasma NFL concentrations was not as accurate because the AUC is 0.630. This suggests that the plasma NFL concentration is not suitable for differential diagnosis between AD and PD/PDD. Alternatively, plasma NFL concentrations could be a promising biomarker for screening for neurodegenerative diseases.

## Discussion

In this work, the reagent for assaying NFL while utilizing the IMR platform showed ultrasensitive detection at a sub-fg/ml concentration. According to published papers, several assay platforms, such as enzyme-linked immunosorbent assay (ELISA) [30–33], single-molecule assay

(a)

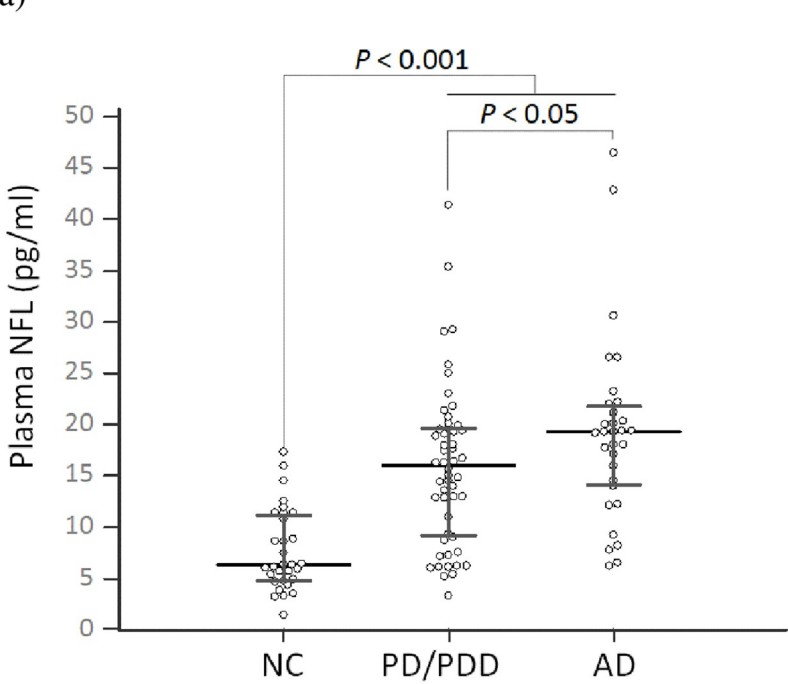

(b)

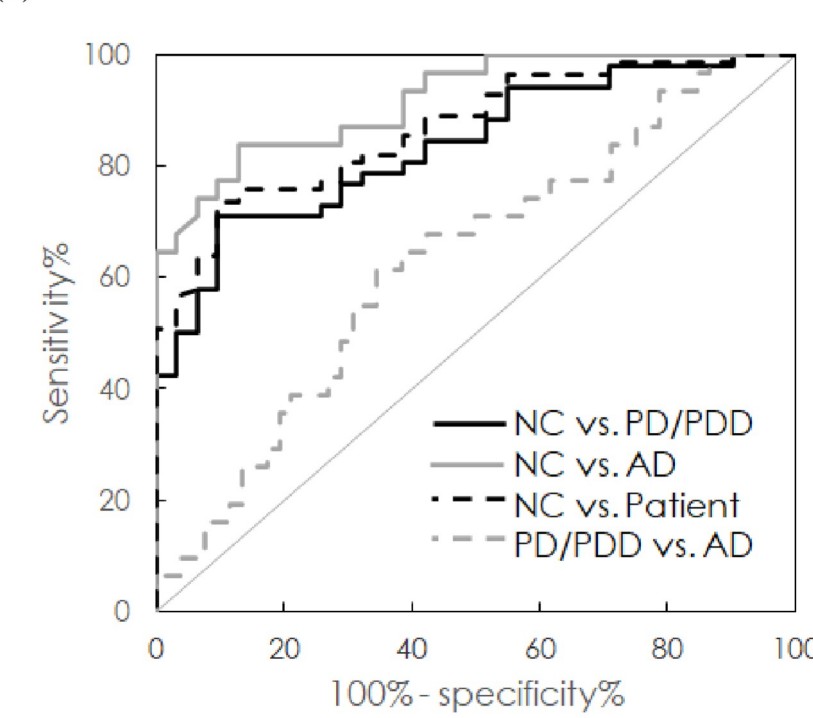

**Fig 3. (A) Measured NFL concentrations in human plasma using IMR and (B) ROC curves used for differentiating various diagnostic groups (NC: Normal control, PD: Parkinson's disease, PDD: Parkinson's disease dementia, AD: Alzheimer's disease, dementia: PD/PDD+AD).** The horizontal line represents median and the range of error bar represents interquartile range.

**Table 11. Results of the ROC curve analysis using plasma NFL concentrations as a discriminative index.**

| ROC curve analysis / Group | Cut-off value of the plasma NFL concentration (pg/ml) | Sensitivity (95% CI) | Specificity (95% CI) | AUC (95% CI) |
|---|---|---|---|---|
| NC vs. PD/PDD | 12.71 | 0.712 (0.569–0.829) | 0.903 (0.743–0.980) | 0.838 (0.754–0.921) |
| NC vs. AD | 12.04 | 0.838 (0.663–0.946) | 0.871 (0.702–0.964) | 0.919 (0.855–0.984) |
| NC vs. Patient* | 12.71 | 0.735 (0.627–0.826) | 0.903 (0.743–0.980) | 0.868 (0.803–0.933) |
| PD/PDD vs. AD | 18.02 | 0.613 (0.422–0.782) | 0.654 (0.509–0.780) | 0.630 (0.507–0.753) |

ROC: receiver operating characteristic AUC: area under the curve

*: including PD/PDD and AD NC: normal control PD: Parkinson's disease

PDD: Parkinson's disease dementia AD: Alzheimer's disease

(SIMOA) [18,34] or electrochemiluminescence (ECL) [35], have been utilized to detect NFL. The reported limits of detection (LoDs) in representative works are listed in Table 13. Depending on the antibodies used, a given assay platform might show various LoDs. Roughly speaking, ECL is more sensitive than ELISA but less sensitive than SIMOA. All three assay platforms show sensitivity at a level of pg/ml or higher in terms of the LoDs for assaying NFL. However, IMR shows a LoD of sub-fg/ml for assaying NFL. This evidence indicates that the IMR assay is much more sensitive than either ELISA, SIMOA or ECL by at least 1000-fold. Notably, the results of interference tests shown in Table 9 reveal the high specificity of assaying NFL with IMR. Thus, the IMR NFL assay is not only ultrasensitive but also specific.

The reasons for achieving ultrasensitive and highly specific IMR assays have been discussed in previous works [36,37]. There are three key factors involved in achieving high sensitivity and specificity for the IMR assay. One is the utilization of antibody-functionalized magnetic nanoparticles homogeneously suspended in liquid. The surfaces of these magnetic nanoparticles serve as substrates of antibodies to catch target antigens. The total area of the association between antibodies and antigens in the IMR assay is obviously larger than that ELISA or ECL.

Another key factor is the use of the ultrasensitive sensor to detect the tiny reductions in magnetic signals due to the associations between antibodies and antigens. The adopted sensor is a high-temperature superconducting quantum interference device (SQUID) magnetometer. The noise level of the SQUID magnetometer is on the order of approximately $fT/\sqrt{Hz}$. The magnetic signal generated by core $Fe_3O_4$ magnetic nanoparticles that are 35 nm in diameter is approximately 1 $pT/\sqrt{Hz}$ [36]. This means that, in principle, the change in the magnetic signal

**Table 12. Clinical sensitivity and specificity of the under the curve (AUC) value for discriminating PD/PDD or AD patients from NCs according to NFL concentrations in blood.**

| Control (n) | Patient (n) | Sample | Sensitivity | Specificity | AUC | Ref. |
|---|---|---|---|---|---|---|
| NC (31) | PD/PDD (52) | Plasma | 0.712 | 0.903 | 0.838 | This work |
| NC (26) | PD (20) | Plasma | 0.82 | 0.92 | - | 16 |
| NC (40) | PD/PDD (116) | Plasma | 0.532 | 0.905 | 0.754 | 21 |
| NC (52) | PD/PDD (139) | Serum | 0.61 | 0.68 | 0.64 | 29 |
| NC (31) | AD (31) | Plasma | 0.838 | 0.871 | 0.919 | This work |
| NC (193) | MCI (197) + AD (180) | Plasma | - | - | 0.87 | 1 |
| NC (41) | aMCI (25) + ADD (33) | Plasma | 0.84 | 0.78 | 0.92 | 20 |

NC: normal control PD: Parkinson's disease

PDD: Parkinson's disease dementia AD: Alzheimer's disease

aMCI: amnesic mild cognitive impairment

**Table 13. Limits of detection (LoDs) reported for various assay platforms, such as enzyme-linked immunosorbent assay (ELISA), electrochemiluminescence (ECL) and single-molecule assay (SIMOA).**

| Assay platform | LoD | Ref. |
| --- | --- | --- |
| ELISA | 15.6 pg/ml | 30 |
| ELISA | 5 pg/ml | 31 |
| ELISA | 125 pg/ml | 32 |
| ELISA | 250 pg/ml | 33 |
| SIMOA | 0.97 pg/ml | 34 |
| SIMOA | 2.2 pg/ml | 18 |
| ECL | 15.6 pg/ml | 35 |
| IMR | 0.18 fg/ml | This work |

due to a single magnetic nanoparticle can be detected. Notably, a sample with an ultralow concentration has an extremely low number of target antigens. After mixing the sample with IMR reagent, only a few magnetic nanoparticles can bind with the target antigen. The reduction in the magnetic signal of the reagent would be very small. As described, the SQUID magnetometer is sensitive enough to detect the tiny reduction in the magnetic signal.

The other factor is the so-called "spin-wash" mechanism involved in IMR [37]. Once a biomolecule binds with the antibody on a magnetic nanoparticle, the bound biomolecule experiences a centrifugal force because the nanoparticle is rotating. The specific biomolecule shows a stronger binding force with the antibody than the nonspecific biomolecule. By controlling the rotating frequency of the nanoparticles adequately, the strength of the centrifugal force can be made stronger than that of nonspecific binding but weaker than that of specific binding. Hence, nonspecific binding is significantly suppressed. Remarkably, the specificity of the antibody against NFL also plays a role in the "spin-wash" mechanism. Therefore, due to these three key factors, IMR shows merits of high sensitivity and high specificity in assaying NFL.

## Conclusion

An assay kit for the measurement of NFL utilizing immunomagnetic reduction was developed. The preclinical characterizations performed according to CLSI guidelines revealed that the IMR NFL assay is ultrasensitive, highly specific and reliable. By applying the IMR NFL kit for assaying plasma NFL in NCs and PD/PDD and AD patients, an accuracy higher than 80% in discriminating PD/PDD or AD patients from NCs was shown. However, the concentrations of plasma NFL are not suitable for differentiating PD/PDD from AD.

## Supporting information

**S1 Data.**
(XLSX)

## Author Contributions

**Conceptualization:** Huei-Chun Liu, Shieh-Yueh Yang.

**Data curation:** Huei-Chun Liu, Chin-Yi Lin.

**Formal analysis:** Huei-Chun Liu.

**Investigation:** Wei-Che Lin, Ming-Jang Chiu, Cheng-Hsien Lu.

**Methodology:** Huei-Chun Liu, Chin-Yi Lin, Shieh-Yueh Yang.

**Writing – original draft:** Shieh-Yueh Yang.

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
