## [Decision Letter · Decision Letter 0]

31 Mar 2020

PONE-D-20-04529

Development of an assay of plasma neurofilament light chain utilizing immunomagnetic reduction technology

PLOS ONE

Dear Dr. Yang,

Thank you for submitting your manuscript to PLOS ONE. After careful consideration, we feel that it has merit but does not fully meet PLOS ONE’s publication criteria as it currently stands. Therefore, we invite you to submit a revised version of the manuscript that addresses the points raised during the review process.

One reviewer raised several concerns on the statistical procedures and their descriptions in the manuscript. The authors should address these issues thoroughly in your responses and the revision of the manuscript.

We would appreciate receiving your revised manuscript by May 15 2020 11:59PM. To enhance the reproducibility of your results, we recommend that if applicable you deposit your laboratory protocols in protocols.io, where a protocol can be assigned its own identifier (DOI) such that it can be cited independently in the future. For instructions see: http://journals.plos.org/plosone/s/submission-guidelines#loc-laboratory-protocols

We look forward to receiving your revised manuscript.

Kind regards,

Kewei Chen, Ph.D

Academic Editor

PLOS ONE

Journal Requirements:

2. Please include your tables as part of your main manuscript and remove the individual files. Please note that supplementary tables (should remain/ be uploaded) as separate "supporting information" files

"Huei-Chun Liu and Shieh-Yueh Yang are employees at MagQu Co., Ltd. Shieh-Yueh Yang is also an employee of MagQu LLC. Shieh-Yueh Yang owns stock in MagQu Co., Ltd. There are no conflicts of interest for other authors. The other authors have no competing interests."

We note that one or more of the authors are employed by a commercial company: MagQu Co., Ltd. and MagQu LLC

Reviewers' comments:

Reviewer's Responses to Questions

**Comments to the Author**

1. Is the manuscript technically sound, and do the data support the conclusions?

Reviewer #1: Yes

Reviewer #2: Yes

2. Has the statistical analysis been performed appropriately and rigorously? 

Reviewer #1: No

Reviewer #2: Yes

3. Have the authors made all data underlying the findings in their manuscript fully available?

Reviewer #1: Yes

Reviewer #2: Yes

4. Is the manuscript presented in an intelligible fashion and written in standard English?

Reviewer #1: Yes

Reviewer #2: Yes

5. Review Comments to the Author

Reviewer #1: This paper details the validation of a novel assay for neurofilament light (NfL), a commonly-used biomarker of neurodegeneration derived from CSF or blood. The authors do a good job of describing the background and importance of this biomarker as it relates to AD research and diagnosis. They also provide a rigorous and comprehensive description of their assay which is helpful in interpreting their results. There are several statistical issues that need to be addressed in this manuscript which are outlined below.

Statistical Methods

1. The authors give a brief description of the statistical analyses that were employed, but they did not describe the analyses used to ascertain the diagnostic accuracy of their NfL assay. They report sensitivity, specificity, and AUC values, but it is unclear what statistical tests were used to derive these values.

2. The authors also state that t-tests were used to compare continuous variables, but it is unclear what comparisons they are referring to. This statement is very vague as it is unclear if the authors are referring to paired t-tests for the reproducibility studies or whether they used a series a two-sample t-tests for the clinical group comparisons. If two-sample t-tests were used for the clinical group comparisons, this would be an incorrect approach. When comparing three or more groups, a one-way ANOVA should be used so that the groups are compared simultaneously and so that the p-values of groupwise comparisons that are statistically significant are adjusted for multiple comparisons.

3. Did the authors check to ensure that the distribution of NfL values met the normality assumption. If they are not normally distributed, then non-parametric tests should be used to test group differences.

4. Why did the authors report the diagnostic accuracy results in the Discussion section? This information should be included in the “Plasma NFL concentrations in patients with dementia” section. In addition, I emphasize the need to describe the statistical methods that were used to derive the AUC values, sensitivity, and specificity. Were these from a logistic regression model or a specific ROC function. Which of the stated statistical software packages were used for these analyses?

5. For AUC, sensitivity, and specificity, please report 95% confidence intervals with these values.

Reviewer #2: An ultrasensitive assay for measuring plasma NFL utilizing IMR technology has been developed. The NC group showed a plasma NFL level of 7.70 ± 4.00 pg/ml, which is significantly lower than that of the PD/PDD (15.85 ± 7.82 pg/ml, p < 0.001) and AD (19.24 ± 8.99 pg/ml, p < 0.001) groups . A significant difference in plasma NFL levels was determined between the PD and AD groups (p < 0.01). Through ROC curve analysis, the cut-off value of the plasma NFL concentration for differentiating NCs from dementia patients (AD and PD/PDD) was found to be 12.71 pg/ml, with a clinical sensitivity and specificity of 73.5% and 90.3% , respectively . The AUC was 0.868. Furthermore, the cut-off value of the plasma NFL concentration for discriminating AD from PD/PDD was found to be 18.02 pg/ml, with a clinical sensitivity and specificity of 61.3% and 65.4%, respectively. The AUC was 0.630. Although clear differences in plasma NFL concentrations were observed among NCs and PD and AD patients, these diagnostic cut-off values should be validated in another group of more patients in further studies.

6. PLOS authors have the option to publish the peer review history of their article (what does this mean?). If published, this will include your full peer review and any attached files.

Reviewer #1: Yes: Michael Malek-Ahmadi

Reviewer #2: No

---

## [Author Response · Author response to Decision Letter 0]

18 May 2020

Comment #1: Please ensure that your manuscript meets PLOS ONE's style requirements, including those for file naming. The PLOS ONE style templates can be found at http://www.plosone.org/attachments/PLOSOne_formatting_sample_main_body.pdf and http://www.plosone.org/attachments/PLOSOne_formatting_sample_title_authors_affiliations.pdf

Response #1: The manuscript has been re-formatted according to the guidelines.

Comment #2: Please include your tables as part of your main manuscript and remove the individual files. Please note that supplementary tables (should remain/ be uploaded) as separate "supporting information" files

Response #2: Tables have been integrated into the main body of the manuscript.

Comment #3: PLOS requires an ORCID iD for the corresponding author in Editorial Manager on papers submitted after December 6th, 2016. Please ensure that you have an ORCID iD and that it is validated in Editorial Manager. To do this, go to ‘Update my Information’ (in the upper left-hand corner of the main menu), and click on the Fetch/Validate link next to the ORCID field. This will take you to the ORCID site and allow you to create a new iD or authenticate a pre-existing iD in Editorial Manager. Please see the following video for instructions on linking an ORCID iD to your Editorial Manager account: https://www.youtube.com/watch?v=_xcclfuvtxQ

Response #3: The corresponding author, Shieh-Yueh Yang, has an ORCID ID: 0000-0002-7850-4597.

Comment #4:　Please provide an amended Funding Statement declaring this commercial affiliation, as well as a statement regarding the Role of Funders in your study. If the funding organization did not play a role in the study design, data collection and analysis, decision to publish, or preparation of the manuscript and only provided financial support in the form of authors' salaries and/or research materials, please review your statements relating to the author contributions, and ensure you have specifically and accurately indicated the role(s) that these authors had in your study. You can update author roles in the Author Contributions section of the online submission form.

Response #4: Done.

Comment #5: Please also include the following statement within your amended Funding Statement. “The funder provided support in the form of salaries for authors [insert relevant initials], but did not have any additional role in the study design, data collection and analysis, decision to publish, or preparation of the manuscript. The specific roles of these authors are articulated in the ‘author contributions’ section.”

Response #5: Done.

Comment #6: Please also provide an updated Competing Interests Statement declaring this commercial affiliation along with any other relevant declarations relating to employment, consultancy, patents, products in development, or marketed products, etc.

Response #6: “MagQu Co., Ltd. and MagQu LLC do not alter our adherence to PLOS ONE policies on sharing data and materials.” is added into the revised manuscript.

Comment #7: Please include both an updated Funding Statement and Competing Interests Statement in your cover letter.

Response #7: Done.

Comment #8: The authors give a brief description of the statistical analyses that were employed, but they did not describe the analyses used to ascertain the diagnostic accuracy of their NFL assay. They report sensitivity, specificity, and AUC values, but it is unclear what statistical tests were used to derive these values.

Response #8: We thank reviewers for your comments. We have re-written the statistics subsection in the revised manuscript. 

In addition, the following sentences are added in the statistics subsection for the tests of sensitivity, specificity and AUC:

“To discriminate PD/PDD (or AD) patients from NCs, receiver operating characteristic (ROC) curve analysis was conducted. The optimal cutoff of NFL concentrations was determined by the Youden index. The confidence interval of area under the curve, sensitivity and specificity was calculated using the DeLong’s nonparametric method.”

Comment #9: The authors also state that t-tests were used to compare continuous variables, but it is unclear what comparisons they are referring to. This statement is very vague as it is unclear if the authors are referring to paired t-tests for the reproducibility studies or whether they used a series a two-sample t-tests for the clinical group comparisons. If two-sample t-tests were used for the clinical group comparisons, this would be an incorrect approach. When comparing three or more groups, a one-way ANOVA should be used so that the groups are compared simultaneously and so that the p-values of groupwise comparisons that are statistically significant are adjusted for multiple comparisons.

Response #9: The following sentences are added in the statistics subsection in the revised manuscript to clarify we were only interested in comparing normal control to disease groups and PD/PDD to AD groups:

“The data of NFL concentrations were checked for normality using the Kolmogorov-Smirnov test and the result showed suggested a violation in the normality assumption (p = 0.033; data not shown). Therefore, the comparison of NFL concentrations between groups (i.e., normal control vs. disease groups; PD/PDD vs. AD groups) was made using the Mann-Whitney U-test.“

The demographic information about Patient, including PD/PDD and AD, is added in Table 10 in the revised manuscript.

Comment #10: Did the authors check to ensure that the distribution of NFL values met the normality assumption? If they are not normally distributed, then non-parametric tests should be used to test group differences.

Response #10: We found the NFL values violate the normality assumption. Therefore, we used Mann-Whitney U-test when comparing groups. The following sentences to illustrate this point are added in Statistics subsection:

“The data of NFL concentrations were checked for normality using the Kolmogorov-Smirnov test and the result showed suggested a violation in the normality assumption (P = 0.033; data not shown). Therefore, the comparison of NFL concentrations between groups (i.e., normal control vs. disease groups; PD/PDD vs. AD groups) was made using the Mann-Whitney U-test.“

Meanwhile, the Figure 3A is also revised.

Comment #11: Why did the authors report the diagnostic accuracy results in the Discussion section? This information should be included in the “Plasma NFL concentrations in patients with dementia” section. In addition, I emphasize the need to describe the statistical methods that were used to derive the AUC values, sensitivity, and specificity. Were these from a logistic regression model or a specific ROC function. Which of the stated statistical software packages were used for these analyses?

Response #11: The diagnostic accuracy results have been moved to “Plasma NFL concentrations in patients with dementia”. The detailed information about statistics is described in Statistics subsection in the revised manuscript.

Comment #12: For AUC, sensitivity, and specificity, please report 95% confidence intervals with these values

Response #12: The values with 95% CI are added in Table 11.

Comment #13: Although clear differences in plasma NFL concentrations were observed among NCs and PD and AD patients, these diagnostic cut-off values should be validated in another group of more patients in further studies

Response #13: Thank for the valuable suggestion. We will do the validation by enrolling independent cohorts in the near future.

---

## [Decision Letter · Decision Letter 1]

28 May 2020

Development of an assay of plasma neurofilament light chain utilizing immunomagnetic reduction technology

PONE-D-20-04529R1

Dear Dr. Yang,

We are pleased to inform you that your manuscript has been judged scientifically suitable for publication and will be formally accepted for publication once it complies with all outstanding technical requirements.

With kind regards,

Kewei Chen, Ph.D

Academic Editor

PLOS ONE

Additional Editor Comments (optional):

Reviewers' comments:

Reviewer's Responses to Questions

**Comments to the Author**

1. If the authors have adequately addressed your comments raised in a previous round of review and you feel that this manuscript is now acceptable for publication, you may indicate that here to bypass the “Comments to the Author” section, enter your conflict of interest statement in the “Confidential to Editor” section, and submit your "Accept" recommendation.

Reviewer #1: All comments have been addressed

2. Is the manuscript technically sound, and do the data support the conclusions?

Reviewer #1: Yes

3. Has the statistical analysis been performed appropriately and rigorously? 

Reviewer #1: Yes

4. Have the authors made all data underlying the findings in their manuscript fully available?

Reviewer #1: Yes

5. Is the manuscript presented in an intelligible fashion and written in standard English?

Reviewer #1: Yes

6. Review Comments to the Author

Reviewer #1: The authors have addressed all of my comments sufficiently and no further revisions are required for this manuscript.

7. PLOS authors have the option to publish the peer review history of their article (what does this mean?). If published, this will include your full peer review and any attached files.

Reviewer #1: Yes: Michael Malek-Ahmadi

---

## [Editor Report · Acceptance letter]

1 Jun 2020

PONE-D-20-04529R1 

Development of an assay of plasma neurofilament light chain utilizing immunomagnetic reduction technology 

Dear Dr. Yang:

I am pleased to inform you that your manuscript has been deemed suitable for publication in PLOS ONE. Congratulations! Your manuscript is now with our production department. 

With kind regards,

on behalf of

Prof. Kewei Chen 

Academic Editor

PLOS ONE